# Effectiveness of an Early Intervention in Mild Hyponatremia to Prevent Accidental Falls in Hospitalized Older Adults—A Crossover Ecological Clinical Trial

**DOI:** 10.3390/healthcare13080865

**Published:** 2025-04-10

**Authors:** Carmen Lobo-Rodríguez, Azucena Pedraz-Marcos, Juan Francisco Velarde-García, Elena Calderari Fernández, Carmen Gadea-Cedenilla, Margarita Medina-Torres, Mª Nieves Moro-Tejedor, Leonor Sánchez García, Ana Mª García-Pozo

**Affiliations:** 1Department of Nursing, Hospital General Universitario Gregorio Marañón, 28007 Madrid, Spain; carmen.lobo@salud.madrid.org (C.L.-R.); elena.calderari@salud.madrid.org (E.C.F.); mmedinat@salud.madrid.org (M.M.-T.); mnieves.moro@salud.madrid.org (M.N.M.-T.); lsanchezgarcia@salud.madrid.org (L.S.G.); agarciapozo@salud.madrid.org (A.M.G.-P.); 2Research Nursing Group of Instituto de Investigación Sanitaria Gregorio Marañón (IiSGM), 28007 Madrid, Spain; 3Department of Nursing, Faculty of Nursing, Physiotherapy and Podiatry, Universidad Complutense de Madrid, 28040 Madrid, Spain; 4Nursing and Healthcare Research Unit (Investén-isciii), Instituto de Salud Carlos III, 28029 Madrid, Spain; azucena.pedraz@isciii.es; 5Research Group in Social Health Care Needs for the Population at Risk of Exclusion, School of Nursing, Red Cross University, Autonomous University of Madrid, 28003 Madrid, Spain; 6Research Group of Humanities and Qualitative Research in Health Science of Universidad Rey Juan Carlos (Hum&QRinHS), 28922 Alcorcón, Spain; 7Nurse for Continuity of Care, Hospital General Universitario Gregorio Marañón, 28007 Madrid, Spain; 8Nursing Research Support Unit, Hospital General Universitario Gregorio Marañón, 28007 Madrid, Spain; 9Department of Nursing, Health Sciences Universidad San Rafael-Nebrija, 28036 Madrid, Spain

**Keywords:** nursing, accidental falls, aged, hyponatremia, patient safety, risk factors

## Abstract

**Background**: Falls in hospitalized patients cause injuries of varying severity and even death. There is a link between falls and low blood sodium levels in older patients. Identifying and treating hyponatremia could help prevent falls and reduce hospital stays. The purpose of this study was to evaluate the effectiveness of the correction of hyponatremia on reducing the incidence of falls and the mean stay of hospitalized patients aged more than 65 years. **Methods**: A crossover ecological clinical trial was conducted in adult hospitalization units of a hospital in Madrid (Spain) over 12 months. Patients meeting inclusion criteria were divided into two randomized groups. The intervention was applied in two six-month phases, alternating between groups with a 15-day washout period. Early diagnosis and treatment of hyponatremia were implemented in the intervention group, while the control group received standard care. Primary outcomes included fall incidence and length of hospital stay. Data were collected using REDCap and analyzed with SPSS v.21. Statistical significance was set at *p* < 0.05 (ClinicalTrials identifier of the manuscript: NCT03265691). **Results**: A total of 1925 patients were included (408 intervention, 1517 control). Fall incidence was significantly lower in the intervention group (6.7 vs. 9.8, *p* = 0.000). Hyponatremia was corrected in 72% of cases. No significant differences were found in functional scores. The intervention effectively reduced falls compared to standard care. **Conclusions**: Early hyponatremia treatment reduces falls and hospital stay in older patients, supporting its inclusion in fall prevention strategies.

## 1. Introduction

Hospitalized patients, especially older adults, are at high risk of falls and serious injuries [1]. The World Health Organization (WHO) defines falls as “an event which results in a person coming to rest inadvertently on the ground or floor or other lower level” [2]. The prevalence of falls in hospitalized patients ranges from 3.56% to 7.3% depending on the study [3,4]. Between 30% and 50% of patients who experience falls in the hospital suffer injuries, such as abrasions and bruises as well as more severe injuries like femur and hip fractures or traumatic brain injury. In the most extreme cases, these falls can even result in the patient’s death [5]. Additionally, patient falls contribute to prolonging the length of hospital stay, resulting in additional care costs and potentially having consequences for the institution’s credibility and legal issues [6]. Falls can be influenced by intrinsic factors, such as advanced age, a history of previous falls, medical conditions, and pharmacological treatments, or by extrinsic factors related to the environment, such as poor lighting, slippery carpets, wet floors, or unexpected obstacles [7]. Hospital fall prevention strategies include patient education, clinician education, environmental adaptations, the use of assistive devices, therapeutic exercises, medication reviews, optimal nutrition, management of cognitive impairment, and falls mitigation policies, systems, and leadership [8,9].

Institutions have implemented various measures to prevent falls; although, these have not proven effective [10]. Despite the implementation of various fall prevention measures, these interventions often face difficulties in achieving the desired outcomes. Recent studies have highlighted the complexity of risk factors, the inadequacy of assessment tools, and the variability of implementation strategies as key obstacles. Multifactorial approaches, personalized interventions, and ongoing staff training have shown promise in reducing fall rates [11,12].

Hyponatremia is a common electrolyte imbalance in hospitalized adults over 65 years old, with a prevalence ranging from 17.7% to 60%, depending on the studied population and the severity of the disorder. Meanwhile, the associated mortality rates in elderly patients reach 17.4%, significantly higher than in patients without hyponatremia [13]. Hyponatremia, which is defined as serum sodium levels < 135 mMol/L, is the most frequent electrolyte disorder [14]. The usual symptoms of acute hyponatremia include lethargy, loss of consciousness, lack of coordination, disorientation, confusion, and unstable gait [15]. The most subtle and nonspecific symptoms are usually observed in chronic hyponatremia and include fatigue, confusion, nausea, dizziness, cognitive impairments, gait disorders, muscle cramps, and a predisposition to falls and fractures. These symptoms are similar to those resulting from moderate alcohol intake, thus explaining the association between hyponatremia and falls and fractures in older adults [4]. The fact that hyponatremia can occur without symptoms, despite serum sodium levels < 120 mMol/L, could be due to adaptation of the brain to hyponatremia [15].

Mild hyponatremia is defined as a serum sodium concentration between 130 and 134 mEq/L. This condition is common and may be associated with mild symptoms such as nausea, vomiting, weakness, headache, and mild neurocognitive deficits. Although mild hyponatremia may be asymptomatic, studies have shown that even mild levels of hyponatremia are associated with increased mortality and morbidity, including an increased risk of falls and fractures, especially in elderly patients. Management of mild hyponatremia usually includes water restriction and regular monitoring of sodium levels, especially in patients with underlying conditions such as cirrhosis [16,17].

The Downton fall risk assessment scale is one of the most widely used scales for measuring the risk of falls [18]. This scale evaluates aspects such as intake of medication, sensory deficit, unstable gait, and previous falls. Some of the drugs covered by this scale are considered inducers of hyponatremia [14]. Furthermore, the symptoms evaluated by the scale as being associated with a risk of falls—sensory deficits, disorientation, unstable gait—could be associated with hyponatremia, thus making the scale a potentially indirect measure of hyponatremia. Therefore, it is necessary to evaluate if the early diagnosis and treatment of hyponatremia have an effect on the incidence of falls and on the length of hospital stay. A positive association between these could lead the intervention to be included in fall prevention protocols throughout healthcare settings.

The aim of this study was to evaluate the effectiveness of mild hyponatremia corrective activity in reducing the incidence of falls in hospitalized patients over 65 years of age.

## 2. Materials and Methods

### 2.1. Design

A crossover ecological clinical trial was conducted over 12 months, with data collection taking place between 2016 and 2017, in the adult hospitalization units of a major hospital in Madrid (Spain) with the highest incidence of falls. The study was initially designed to be conducted in the Neurology, Rehabilitation, Geriatrics, Internal Medicine, Cardiology, Gastroenterology, Neurosurgery, and Oncology units, as specified in the protocol. However, to enhance the study’s comprehensiveness and better address the specific needs of the elderly population, Ortho-Geriatrics and Geriatric Convalescence units were also included. These units primarily care for patients from the originally specified specialties, ensuring methodological consistency while broadening the scope of the findings. The inclusion of these additional settings adhered strictly to the study’s original methodology, preserving the reliability and validity of the results. The ClinicalTrials identifier of the manuscript is NCT03265691.

### 2.2. Inclusion/Exclusion Criteria and Samples

The study included all patients admitted to the aforementioned hospitalization units during the study period. Patients were included if they met the following criteria: (1) conscious and oriented; (2) not in pain or suffering from a terminal illness; (3) not bedridden and able to walk; and (4) willing to participate in the study with signed informed consent. Conversely, patients were excluded if they (1) were transferred to other units not included in the study during the follow-up period; (2) experienced a deterioration in their condition; or (3) had severe impairment of their functional and/or psychological capacity.

### 2.3. Intervention Design and Implementation

The intervention was conducted on hospitalized patients, while the outcome variables were fall indicators and overall mean length of stay in the study units. The trial was performed based on 2 groups (A and B). Assignment of the units to the groups and the group that was to start with the intervention during the first phase were randomized. Study was carried out in 2 phases, each lasting 6 months, as follows:

First, the intervention was carried out in Group A. Group B was the control group, in which all admitted patients were managed according to usual practice.

Second, the intervention was carried out in Group B, with Group A as the control group.

In the intervention group, hyponatremia was assessed upon admission and monitored daily until levels normalized.

A washout period of 15 days, longer than the overall mean stay for these units (6.9 days), was left between each phase of the study, ensuring that the participants in the first phase were not included in the second phase during the same admission.

In the intervention group, chronic hyponatremia is usually asymptomatic; therefore, corrective treatment is only applied in daily clinical practice in the most severe cases or in those with frank symptoms of hyponatremia. In this study, corrective treatment was initiated both in patients with clear symptoms of hyponatremia and in those with no symptoms. Hyponatremia (<135 mMol/L) was diagnosed and treated early in the intervention group in both phases. Serum sodium values were monitored at admission and throughout hospitalization. If hyponatremia was detected, medical treatment and corrective measures started. Once the patient was diagnosed with hyponatremia, he/she was followed until serum sodium values returned to normal or discharge from hospital (Figure 1).

The treatment was tailored to the etiology of hyponatremia and the patient’s clinical characteristics and ranged: vasopressor receptor 2 antagonists, discontinuation of hyponatremia-inducing drugs, restriction of intravenous serum therapy, dietary measures, and eliminating salt from the patient’s diet. Monitoring of serum sodium levels, observation of the signs and symptoms of hyponatremia, and informing the patient and/or family members on practical measures for treatment of hyponatremia and on the signs and symptoms of hyponatremia so that they were able to recognize them were performed by the nursing team. In the control group, the patients were managed according to usual practice on admission to the relevant unit.

### 2.4. Follow-Up and Data Collection

The variables common to group intervention and control were hospitalization unit, age, sex, and score on the Downton scale at admission. In the intervention group, we also measured sodium level at admission and during the stay. In addition, if hyponatremia was detected, the pathologic sodium value, date of detection of hyponatremia, date of normalization of blood sodium, and scores on the Functional Ambulation Category (FAC) scale [18], Barthel Scale (BS) [19], and the Short Physical Performance Battery (SPPB) [20] were measured. The FAC is a measurement tool that assesses the need for assistance with ambulation and classifies patients into six possible categories: from 0, where the patient has effective ambulation, to 5 levels of ambulation, depending on the greater or lesser need for physical assistance, supervision by another person, or the level of dependence on walking surface. The use of this scale as an assessment tool allows objective quantification of the patient’s degree of disability [18]; while the BS assesses the degree of a person’s physical functional dependence or the need for help to perform ten basic activities of daily living, which are evaluated with a score of 10, 5, or 0, depending on the degree of help needed (none, some, or all) [19]. Finally, the SPPB assesses lower extremity physical function in older adults, using three main tests of balance, gait speed, and chair raise. Each is scored from 0 to 4, and the sum of the scores of the three tests gives a total score from 0 to 12, where 0 indicates the worst performance and 12 the best [20].

Other variables recorded with hyponatremia and when sodium levels returned to normal included date of discharge, occurrence of hyponatremia between normalization and discharge, blood sodium level if high, initiation of treatment to correct hyponatremia, and correction of sodium values during hospitalization.

At the end of each phase of the study, we collected grouped data from both the intervention and the control patients: overall number of falls, unit where the falls were recorded, and sex of patient who had fallen. As primary outcomes, we evaluated the incidence of falls in both groups at the end of follow-up and the length of stay in the hospital. Study data were collected and managed using REDCap electronic data capture tools hosted at the hospital [21]. REDCap (Research Electronic Data Capture) is a secure, web-based application designed for data capture in research studies.

### 2.5. Statistical Analysis

Statistical analysis was performed using Access and SPSS V.21 for Windows. Hyponatremia was defined as serum sodium values of <135 mMol/L [14]. We performed a univariate analysis overall and stratified according to the intervention and control groups. A possible carry-over effect was evaluated, as this was a crossover clinical trial. The results for falls in the second-phase intervention group were compared with those of the first-phase intervention group [22]. The test was used to determine whether there was an association with the outcome variable “incidence of falls”. A bivariate analysis was performed in order to evaluate homogeneity in the intervention and control groups. Quantitative variables were expressed as measures of central tendency (mean and median) and dispersion (standard deviation [SD] and interquartile range [IQR]), depending on their distribution. Qualitative variables were expressed as absolute frequencies and percentages.

Specific tests were performed to explore the association between the variables: ×2 for qualitative variables, and t test for quantitative variables and 2-factor qualitative variables. The efficacy of the early intervention was measured based on the excess incidence between both groups. Statistical significance was set at *p* < 0.05.

### 2.6. Ethical Considerations

The study has been approved by the Clinical Investigation Ethics Committee of the hospital ‘REDACTED’ where the research was conducted (‘REDACTED’). Participation in the study was voluntary and anonymous. All patients (or a relative or legal guardian) gave their written informed consent for participation. Informed consent was obtained in accordance with the World Medical Association Declaration of Helsinki. Confidentiality was guaranteed at all times.

## 3. Results

The total number of participants was 1925, of whom 408 were in the intervention group and 1517 in the control group. The intervention group included 215 women (52.8%). Mean (SD) age was 82.35 (7.38) years. According to the Downton scale, 94.1% were high risk (n = 384). A score of ≥3 is considered to indicate that the patient is at risk of falling. A total of 28 patients had hyponatremia in the intervention group. Three decided not to continue in the study. Therefore, 25 patients (6.1%) underwent the intervention. Blood sodium levels returned to normal in 18 patients (72.0%). Between correction of hyponatremia and discharge, hyponatremia reappeared in eight patients (44.4%). No statistically significant differences in age or scores (FAC, BS, SPPB) were observed between patients with hyponatremia and those whose blood sodium levels returned to normal. The overall length of stay was 4.04 days shorter in the group whose blood sodium returned to normal, although the difference was not significant (*p* = 0.216) (Table 1).

Of the 1517 participants in the control group, 825 (54%) were women and 692 (46%) were men. The mean age was 81.23 (8.62) years. According to the Downton scale, 92.3% had a high risk of falls. Statistically significant differences were found in both groups for age (*p* = 0.009). In contrast, both groups were homogeneous in terms of sex (*p* = 0.576), and for risk of falls according to the Downton scale (*p* = 0.237), both groups had a high risk.

Statistically significant differences between the groups were found for the number of falls during the study period (*p* = 0.000) and the incidence of falls (*p* = 0.000). The mean incidence of falls was 6.7 (3.1) in the intervention group and 9.8 (4.7) in the control group (Table 2).

The intervention group included all patients who underwent the intervention both in the first and in the second phase. Similarly, the control group included all patients who were managed according to usual clinical practice in both phases of the study. The carry-over effect was analyzed using the t test for paired data. Without taking into account the order of treatment, the mean incidence of falls in the intervention group was lower than in the control group (6.7 [3.1] vs. 9.8 [4.7]); SMD: 0.775.

A comparison of the intervention group in the first phase with the intervention group in the second phase did not reveal statistically significant differences for variables: treatment started (*p* = 0.283), correction of blood sodium during admission (*p* = 0.270), hyponatremia from correction to discharge (*p* = 0.250), and risk of falls according to the Downton scale (*p* = 0.143). Therefore, with respect to these variables, both groups were homogeneous. In contrast, significant differences were found between the groups for sex (*p* = 0.037) (Table 3).

A comparison of the scores on the various scales in patients with hyponatremia for the first- and second-phase intervention groups did not reveal significant differences for the FAC, Barthel Scale, or SPPB scores. The scores on these scales were also compared in patients who underwent the intervention after correction of hyponatremia in both intervention groups. In order to compare the effectiveness of the intervention with respect to daily practice, the overall number of falls was analyzed in the intervention and control groups. Both the absolute number of falls and the incidence of falls were significantly greater in the control group (*p* = 0.000) (Table 4)

## 4. Discussion

In this study, the mean age of the study patients was high (82.35 [7.38] years). Studies show that persons over 80 years of age have the greatest risk of experiencing a fall [2,23]. This greater susceptibility of older adults to hyponatremia is due to, among others causes, polymedication and a greater incidence of diseases [14]. This electrolyte disorder may also be associated with the restriction of salt in the patient’s diet for control of arterial hypertension, however, the effectiveness of this measure has been called into question [24,25]. According to the Downton scale, 94.1% of patients had a high risk of falls (5.2 [1.8]). This scale evaluates previous falls, sensory deficit, mental status, degree of dependence for walking, and intake of drugs such as tranquilizers/sedatives, diuretics, antihypertensive drugs, antidepressants, and other medications [26]. Some of these agents are known to induce hyponatremia; others affect the ability to walk or the level of consciousness. They are all considered to be risk factors according to the Downton scale [25]. Therefore, hyponatremia can be considered an independent risk factor for falls, as shown elsewhere [14,27].

Analysis of the homogeneity of the intervention and control groups revealed differences between the mean number and the incidence of falls (*p* = 0.000), which were lower in the intervention group. As shown in the literature, hyponatremia could predispose falls [15,28,29]. When the first- and second-phase intervention groups were compared to analyze whether the order of intervention affected the study outcomes, they were observed to be homogeneous with respect to treatment started, correction of sodium values during admission, reappearance of hyponatremia between normalization and discharge, sodium values at discharge, and risk of falls according to the Downton scale. In contrast, both the mean number of falls that occurred and the incidence of falls were greater in the second-phase intervention group, thus enabling us to state that the order of intervention did not affect the outcome of the study. The same can be said of length of stay, which was significantly greater (*p* = 0.005) in the second-phase intervention group, with a difference of 3.26 days [24].

Therefore, it is important to correct hyponatremia when it is detected and to follow up with the patient if we want to avoid adverse effects [23]. That shows the importance of continuous care at both specialized and primary care levels. Hyponatremia, even “asymptomatic” hyponatremia, should be corrected, and the withdrawal of drugs that have the potential to induce hyponatremia should be considered [30]. Recent studies show the poor sensitivity of the Downton scale and the STRATIFY tool for detecting the risk of falls [31]. In addition, they show that the functional impairment associated with hospitalization in patients older than 65 years is related to falls and age [32]. Therefore, it is essential to find strategies and scales that act as reliable tools for the prevention of falls in older adults who are the most vulnerable to them. This would help improve the quality of life of older adults and safety care, key elements of quality of care [33].

As for the study design—crossover ecological clinical trial—we were unable to find studies on risk factors for falls with this type of methodological design. As this was an ecological study, and despite the fact that the participants received treatment, we measured behavior in relation to falls in the units, that is, the number and incidence of falls and length of stay in the units. Hospitalization units act as their own control groups. Key aspects of the performance of this type of study are random assignment of patients to each group, evaluation of the effects of the order in which the treatments are received, and the performance of statistical analyses to determine the carry-over effect [22]. In our study, the units were randomly assigned to each study group, and the initial phase in each group was randomized. We then used the t test to compare the means of the sums in the two sequences of order of treatment of hyponatremia (first- and second-phase intervention groups), thus ruling out the carry-over effect. A washout period between the phases is also essential. We chose a period that was longer than the mean stays to ensure that participants from the first phase did not participate again in the second.

### Limitations

The main limitation of the study lies in the use of aggregated data for the outcome variable, which may lead to what is known as the “ecological fallacy”—a misinterpretation that arises when group-level data are generalized to the individual level. This, in turn, can result in erroneous conclusions about associations between events that do not truly exist, with the primary challenge being the control of confounding factors. Despite the limitations inherent to this type of study, its important guiding and preventive value should not be dismissed, although categorical conclusions cannot be drawn. Thanks to its ecological design, this study helps mitigate inter-individual variability, as each hospital unit acts as its own control. However, we acknowledge that this design alone does not fully eliminate the influence of confounding variables. Specifically, bivariate and stratified analyses were conducted to assess homogeneity between groups and control for known confounders such as age, sex, and baseline fall risk (Downton scale). Nevertheless, a multivariate approach (e.g., logistic regression) could have provided more robust control of potential confounding factors. While stratified bivariate analyses were performed, future studies should consider the use of multivariate models to adjust for other relevant confounders, such as comorbidities and medication use, which may influence both hyponatremia and fall risk. Implementing these additional adjustments would enhance the precision and reliability of the findings, offering a more comprehensive understanding of the intervention’s effectiveness.

Another limitation of our study is the absence of the Dynamic Gait Index (DGI). Although the DGI provides valuable information about gait biomechanics, its high cost and the need for specialized equipment limit its feasibility in our clinical setting. Instead, we used validated alternative instruments such as the Functional Ambulation Category (FAC) scale and the Short Physical Performance Battery (SPPB) scale. However, these tools lack the specificity and sensitivity of the DGI for assessing dynamic gait adaptations and balance during complex tasks, such as turning, pivoting, or walking under dual-task conditions. The omission of the DGI represents a deviation from the original protocol and limits our ability to directly capture gait alterations associated with fall risk. Consequently, subtle gait changes following hyponatremia correction may not have been fully detected, potentially leading to an underestimation of the intervention’s effect. Despite its utility in biomechanical analysis, the DGI’s omission could be justified by the patients’ average Barthel Scale scores, which indicate a moderate level of dependence and physical limitations. These patients require significant assistance for certain daily activities, which may hinder their participation in gait and balance assessments. Instead, we prioritized personalized rehabilitation approaches tailored to their needs and abilities, ensuring safety and promoting effective recovery. We strongly recommend that future studies incorporate the DGI or other dynamic assessment tools when resources and context allow. This would enhance the accuracy of fall risk evaluation and help clarify the biomechanical mechanisms through which mild hyponatremia contributes to postural instability and falls in older adults.

The imbalance between the intervention and control groups is a recognized limitation of our study. This discrepancy resulted from the decision to include all eligible hospitalized patients aged 65 years or older during the study period, rather than limiting recruitment to a predefined target. While the increase in the size of the control group enhances statistical power, this imbalance can influence the estimates of the intervention’s effectiveness. In randomized clinical trials, a moderate imbalance typically does not significantly reduce statistical power. However, substantial imbalance can affect the precision and robustness of the results. In our case, the smaller intervention group may be more susceptible to random variability and the influence of outliers, which could distort means and reduce the stability of the findings. Additionally, this imbalance may limit the generalizability of the results, as a larger volume of data in one group does not necessarily compensate for differences in representativeness or comparability between study conditions. Therefore, it is important to consider this imbalance when interpreting the magnitude of the intervention’s effect on fall reduction. We recommend that future research employs more balanced randomization strategies or statistical weighting methods to ensure a more homogeneous distribution of participants across groups, while preserving the real-world applicability.

## 5. Conclusions

While the evaluation of the risk of falls is homogeneous in the intervention and control groups, both the total number of falls and the incidence were higher in the control group. Therefore, according to the data, early diagnosis and treatment of hyponatremia reduce falls in hospitalized patients aged >65 years.

Given that early treatment and intervention for hyponatremia are associated with a reduction in the incidence of falls for older adults in hospital and in length of stay, the monitoring of serum sodium levels should be included in strategies for preventing falls and in scales for measuring the risk of falls. This should be on a widespread basis at all care levels and not only in the hospital setting. Thus, the number of falls outside the hospital will also be reduced.

## Figures and Tables

**Figure 1 healthcare-13-00865-f001:**
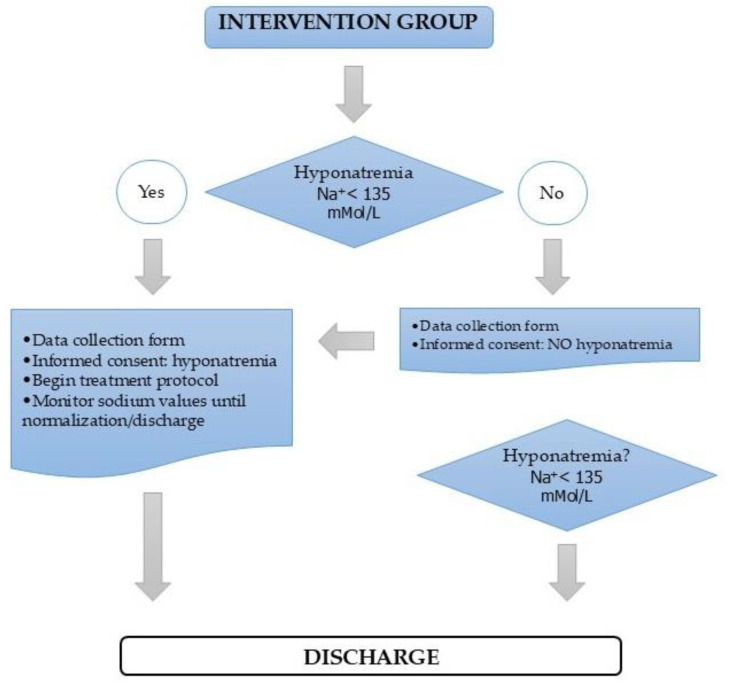
Intervention algorithm.

**Table 1 healthcare-13-00865-t001:** Characteristics of patients who experienced hyponatremia in the intervention group.

Intervention Group	Hyponatremia n = 25	Normalized Blood Sodium n = 18	*p* Value
Age (years) Mean (SD)	83.52 (6.5)	82.28 (7.4)	0.417
Length of stay (days) Mean (SD)	15.56 (15.72)	11.52 (9.64)	0.216
	**Hyponatremia n = 25 (*)**	**Normalized Blood Sodium n = 18**	
**FAC Scale**			
Nonfunctional n (%)	5 (18.5)	3 (12.5)	0.135
Dependent n (%)	18 (66.7)	15 (62.5)	
Independent n (%)	4 (14.8)	6 (25.0)	
**B Scale**			
Unable n (%)	6 (21.4)	3 (12.2)	0.607
Needs major help n (%)	13 (46.4)	13 (52.0)	
Needs some help n (%)	8 (28.6)	8 (32.0)	
Needs little help n (%)	1 (3.6)	1 (4.0)	
**SPPB Scale**			
Frail n (%)	19 (86.4)	18 (85.7)	1
No change n (%)	3 (13.6)	3 (14.3)	

Statistical significance was set at *p* < 0.05. SD, standard deviation. (*) A total of twenty-eight patients had hyponatremia. Of these, three rejected treatment after application of the scales.

**Table 2 healthcare-13-00865-t002:** Analysis of the homogeneity of the intervention and control group.

		CG n = 1517	IG n = 408	*p* Value	Difference	95% CI
Age (years), Mean (SD)		81.23 (8.6)	82.35 (7.4)	0.009	−(1.1)	(−1.96 to 0.82)
Downton scale, Mean (SD)		5.17 (1.9)	5.2 (1.8)	0.665		
Falls, Mean (SD)		18.56 (7.21)	17.05 (6.65)	0.000	1.51	(0.74–2.299)
Incidence of falls, Mean (SD)		9.80 (4.7)	6.70 (3.1)	0.000	3.10	(2.71–3.49)
Sex	Male, No (%)	692 (45.62)	192(47.06)	0.576		
	Female, No (%)	825 (54.38)	215 (52.94)			
Risk of falls on Downton scales	High, No (%)	1355 (89.32)	384 (94.12)	0.237		
	Low, No (%)	113 (10.68)	24 (5.88)			

Statistical significance was set at *p* < 0.05. SD, standard deviation. Control group (CG). Intervention group (IG).

**Table 3 healthcare-13-00865-t003:** Analysis of the homogeneity of the first- and second-phase intervention groups.

		Second-Phase Group	First-Phase Group	*p* Value
Sex, No (%)	Male	56 (56.56)	136 (44.15)	0.037
	Female	43 (43.44)	172 (55.85)	
Treatment started, No (%)	Yes	5 (100)	16 (66.66)	0.283
	No	0 (0)	8 (33.34)	
Na+ corrected, No (%)	Yes	1(33.3)	12 (70.58)	0.270
	No	2 (66.7)	5 (29.42)	
Hyponatremia between normalization and discharge, No (%)	Yes	1 (100)	3 (20)	0.250
	No	0 (0)	12 (80)	
Risk of falls according to Downton scale, No (%)	Low	9 (9.00)	15 (4.88)	0.143
	High	91 (91.00)	293 (95.12)	

Comparison between the second-phase intervention group and the first-phase intervention group. Statistical significance was set at *p* < 0.05.

**Table 4 healthcare-13-00865-t004:** Comparison of the first- and second-phase intervention groups.

	Second-Phase Group (n = 100)	First-Phase Group (n = 308)	*p* Value	Mean Difference	95% CI
Age, mean (SD)	79.61 (6.1)	83.24 (7.53)	0.000	3.63	(2.00–5.27)
Downton score, mean (SD)	4.91 (1.87)	5.32 (1.87)	0.059		
Hyponatremia, mean (SD)	131.40 (3.20)	130.20 (3.24)	0.461		
FAC Scale Hyponatremia, mean (SD)	3.40 (0.89)	1.55 (1.33)	0.007	1.85	(0.55–3.16)
Barthel Hyponatremia, mean (SD)	68.00 (26.12)	45.7 (25.8)	0.093		
SPPB Hyponatremia, mean (SD)	3.18 (5.27)	3.18 (4.14)	0.141		
Nº of falls, mean (SD)	15.18 (8.58)	17.66 (5.78)	0.008	2.48	(0.66–4.29)
Incidence of falls, mean (SD)	10.65 (3.90)	5.41 (1.19)	0.000	5.24	(4.45/6.02)
Blood sodium at discharge, mean (SD)	139.07 (3.49)	138.74 (3.57)	0.428		

Statistical significance was set at *p* < 0.05. SD, standard deviation.

## Data Availability

The dataset used is restricted and stored at the hospital where the study was conducted and can be obtained from the author upon reasonable request.

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
