# Peer review of "Effectiveness of an Early Intervention in Mild Hyponatremia to Prevent Accidental Falls in Hospitalized Older Adults—A Crossover Ecological Clinical Trial"

_healthcare, 2025, doi:10.3390/healthcare13080865_

Round 1
Reviewer 1 Report
Comments and Suggestions for Authors
Dear Authors,
Your study addresses a highly relevant topic in patient safety and hospital care quality. The crossover ecological trial design is innovative and contributes to the literature on fall prevention strategies. However, there are substantial methodological deviations that prevent alignment with the original registered protocol. Below, I outline the key concerns:
- Hyponatremia is not included as part of routine hospital admission screening. Why? This issue raises ethical concerns. The absence of institutional ethical approval further complicates the study’s validity and adherence to research standards.
- The registered clinical trial protocol (https://clinicaltrials.gov/study/NCT03265691?term=%20NCT03265691&rank=1) was last updated in 2017. This represents a considerable gap between the original study design and the current manuscript. Given the rapid advancements in clinical research, a protocol that has not been updated or verified in nearly eight years raises concerns about its relevance and applicability. The absence of confirmation regarding adherence to the original study plan weakens the validity of the findings. Without a clear explanation of how the study was conducted in accordance with the initial protocol, the manuscript cannot be considered methodologically sound.
- The protocol specifies early detection and treatment of hyponatremia as the experimental intervention, assessed using the Dynamic Gait Index test. However, the manuscript does not report the use of this test, which was a primary component of the intervention evaluation. The omission of key methodological elements affects the study’s transparency and reproducibility.
- A secondary outcome in the protocol was the cost-effectiveness analysis of the two strategies, but this aspect is not discussed in the manuscript. The absence of this evaluation leaves a critical gap in the study’s completeness and alignment with its original objectives.
- Another major limitation is the reliance on outdated references. Seventeen out of twenty references are older than five years. Given the continuous evolution of fall prevention strategies and electrolyte management, it is essential to support the study with recent literature. Additionally, the manuscript does not adhere to the MDPI formatting style, which is a fundamental requirement for scientific rigor.
- The abstract lacks clarity and specificity. The statement "Hospitalized patients are at risk of falls" is too generic and does not adequately focus on the relationship between falls and hyponatremia. The study period is not specified, as "12 months" is too vague. If the primary outcome is length of stay, it should be clearly introduced in the title, background, results, and conclusions. However, the trial does not formally assess length of stay as a primary outcome, and it should not be presented as such to ensure methodological accuracy.
- There is a discrepancy between the study settings described in the manuscript and those listed in the protocol. The protocol mentions that the study was conducted in Neurology, Rehabilitation, Geriatrics, Internal Medicine, Cardiology, Gastroenterology, Neurosurgery, and Oncology units. However, the manuscript includes additional settings such as Ortho-Geriatrics and Geriatric Convalescence. Conducting the study in settings not originally specified introduces a methodological inconsistency.
- A significant issue is the discrepancy between the estimated sample size and the actual number of enrolled participants. The protocol estimated 124 participants per group, for a total of 248, while the manuscript reports 1925 participants, with 408 in the intervention group and 1517 in the control group. This large deviation raises concerns about selection bias and the validity of the statistical analysis. The imbalance between groups further compromises comparability and increases the risk of misleading conclusions. Unfortunately, this issue cannot be resolved without reanalyzing the data from the beginning.
These methodological inconsistencies and deviations from the registered protocol put at risk the scientific rigor of your work.
Comments on the Quality of English LanguageAn academic review is necessary to ensure a scientific sound.
Author Response
Response letter added

Reviewer 2 Report
Comments and Suggestions for Authors
Introduction
The association of falls with hyponatremia, as well as a detailed description of the symptoms and consequences of this electrolyte disorder, offers a deeper understanding of the mechanisms that may contribute to the increased risk of falls.
As an introduction to the topic, the text is informative, well-structured, and provides a solid basis for understanding the problem.
Methodology
The methodology is exciting and complex and seems quite correctly described. However, I do not have enough experience with this type of research, so I cannot say whether it meets all the criteria of scientific methodological research.
Results
The results present a large amount of data obtained clearly and concisely. The results are presented in tabular and graphical form with adequate explanations. The results are interpreted professionally and logically.
The table title should go above the table, not below it as it currently stands.
Discussion
The authors clearly explain how hyponatremia, even in asymptomatic form, can increase the risk of falls, which is very important for clinicians in order to recognize and treat this condition in a timely.
The limitations of the study are listed in the section where the authors state the key weaknesses of the research, which speaks in favor of the fact that the authors are pretty clear that longitudinal research is needed to ensure long-term results and improve tools for the assessment and prevention of falls, as well as treatment strategies for hyponatremia in the elderly.
The conclusions are clearly and precisely formed based on the results and discussion. The conclusions can be reliable and correspond to the set goals.
In the Literature chapter, the literature data used are valid and follow the paper's overall structure. The selection of references could be more current; 1 paper is from 2022, 1 from 2021, and 4 papers from 2018, and the remaining references are significantly older.
Author Response
Response letter added

Reviewer 3 Report
Comments and Suggestions for Authors
The issue of this study is quite interesting, however, some points needed to be improved and clarified. Please see my comments below…
ABSTRACT
“(ClinicalTrials.gov identifier of the manuscript: NCT03265691).” - ??????
“No significant differences were found in functional scores.” – Which functional scores?
INTRODUCTION
P2L50 - “The consequences of a fall vary from mild to very severe lesions, and the fall may even be fatal. Falls are the second leading cause of death from unintentional injury [2].” – In hospital context... This point must be clarified…
P2L56 - “Institutions have implemented various measures to prevent falls, although these have not proven effective [7].” – Which measure?
Data on the incidence of hyponatremia in the hospital setting were important in the Introduction…
METHODS
It is not clear what the time interval was between monitoring the serum sodium...
P3L132 - “If hyponatremia was detected: the pathologic sodium value, date of detection of hyponatremia, date of normalization of blood sodium, score on the Functional Ambulation Categories (FAC) scale, [12], Barthel scale [13] and the Short Physical Performance Battery (SPPB) [14].” - The importance of these tests is not perceived either in the Introduction or in the Materials and Methods section.
Author Response
Response letter added

Round 2
Reviewer 3 Report
Comments and Suggestions for Authors
None.
Author Response
Thank you